# Learning Model Predictive Controllers with Real-Time Attention for Real-World Navigation

**Xuesu Xiao**[*1,2], **Tingnan Zhang**[*3], **Krzysztof Choromanski**[*3], **Edward Lee**[3],
**Anthony Francis**[3], **Jake Varley**[3], **Stephen Tu**[3], **Sumeet Singh**[3], **Peng Xu**[3],
**Fei Xia**[3], **Sven Mikael Persson**[2], **Dmitry Kalashnikov**[3], **Leila Takayama**[4],
**Roy Frostig**[3], **Jie Tan**[3], **Carolina Parada**[3], **and Vikas Sindhwani**[3]
[*]Equally Contributing Authors [1]George Mason University
[2]Everyday Robots [3]Robotics@Google [4]Hoku Labs

**Abstract:** Despite decades of research, existing navigation systems still face real-world challenges when deployed in the wild, e.g., in cluttered home environments or in human-occupied public spaces. To address this, we present a new class of implicit control policies combining the benefits of imitation learning with the robust handling of system constraints from Model Predictive Control (MPC). Our approach, called Performer-MPC,[1] uses a learned cost function parameterized by vision context embeddings provided by Performers—a low-rank implicit-attention Transformer. We jointly train the cost function and construct the controller relying on it, effectively solving end-to-end the corresponding bi-level optimization problem. We show that the resulting policy improves standard MPC performance by leveraging a few expert demonstrations of the desired navigation behavior in different challenging real-world scenarios. Compared with a standard MPC policy, Performer-MPC achieves >**40%** better goal reached in cluttered environments and >**65%** better on social metrics when navigating around humans.

**Keywords:** Model Predictive Control, Transformers, Performers, Highly-Constrained Navigation, Social Navigation, Learning-based Control

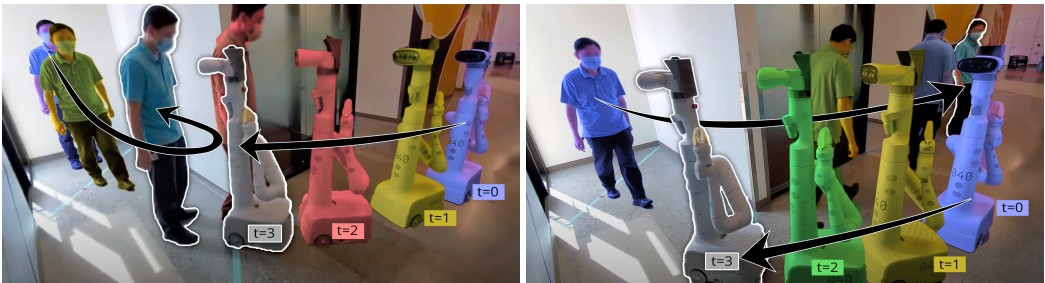

Figure 1: Left: Standard MPC efficiently cuts blind corners, forcing the human to back up; Right: Social Performer-MPC avoids cutting blind corners, enabling safe navigation around humans.

## 1 Introduction and Related Work

Real-world robot deployment in human-centric environments, such as cluttered homes or crowded offices, remains an unsolved problem [1, 2]. These challenging situations require safe and efficient navigation through tight spaces, such as squeezing between coffee tables and couches, handling tight corners, doorways, untidy rooms, and so on. An equally critical requirement is to navigate in a manner that complies with unwritten social norms around humans.

Classical approaches using model-based control [3, 4, 5, 6] can already move robots from one point to another safely and reliably. However, when deploying these systems in the complex real world [7, 8, 9], extensive engineering effort is required to construct world representations [10, 11, 12], model vehicle kinodynamics [13, 14], hand-craft cost functions [15, 16], fine-tune system parameters [17],

---

[1]Project page: `https://performermpc.github.io`

6th Conference on Robot Learning (CoRL 2022), Auckland, New Zealand.

and design backup planners to recover from stuck scenarios [18, 19, 20, 21]. While providing verifiable guarantees, these cascaded components need to be hand-engineered before deployment based on the roboticist's best expectations of what would be encountered in the real world, and become cumbersome when in-situ modifications are necessary to enable adaptive behaviors [17].

In contrast to these classical methods, machine learning enables robots to *learn* these behaviors directly from data [22]. End-to-end learning [23, 24] is an appealing paradigm to reduce the engineering effort and cascading errors caused by separate components, but it usually requires extensive real-world training data or simulation with inevitably simplified human and environment representations. Most importantly, it lacks safety, optimality, generalizability, interpretability, and explainability, which are crucial for real robots moving around humans [22, 25, 26]. Therefore, researchers have looked at individually learning global planners [27], local planners [28, 29, 30, 31, 32, 33], and other navigation components including cost representations [34, 35, 36, 37], kinodynamic models [38, 39], and planner parameters [40, 41, 42, 43, 44, 45, 46] to enable both better navigation performance, and also off-road [47, 48] and social navigation [1, 49, 50].

Both classical and learning-based methods have their merits. Model Predictive Control (MPC) [3, 4, 5, 6] enables synthesis of real-time feedback controllers for robots operating in real-world environments that satisfy given safety constraints, optimality criteria, and kinodynamic models. To get the best of both worlds, we design a class of *Learnable-MPC* policies enabling robots to learn navigation behaviors in real-world use cases by combining the flexibility of learning from demonstrations with the optimality properties (e.g., collision-free, shortest path) of MPC solutions. Our framework can also been seen as a class of Implicit Behavior Cloning policies [51] that are aware of real-world robot-environment and robot-human interactions. Our contributions in this paper are three-fold:

- We augment the cost function of MPC with learnable components parameterized by rich Transformer-based latent embeddings of real-world context. Transformer architectures [52, 53, 54] have produced stunning advances in language modeling [55, 56, 57, 58, 59, 60], image generation [61, 62, 63, 64, 65, 66], and multi-modal reasoning [67, 68, 69, 70, 71, 72]. This indisputable success comes at a computational price in proportion to the massive number of parameters learned (e.g., 175 billion for GPT-3 [56]) as well as quadratic scaling in input sequence length of the core attention modules of these models. By generating context-dependent quadratic costs using Performers [73]—a low-rank linear-attention Transformer, we demonstrate how we can embed powerful *pixel-to-pixel* attention mechanisms in MPC while crucially retaining real-time solutions on a CPU onboard a mobile robot.
- Using distributed bilevel optimization with implicit differentiation mechanisms, we train navigation policies on expert demonstrations to handle difficult navigation scenarios, with data augmentation strategies to mitigate well-known distribution shift issues that frequently plague behavioral cloning and other imitation learning approaches.
- We demonstrate that our Performer-MPC outperforms its counterparts in real-world challenging navigation scenarios, including highly constrained and human-occupied environments. Performer-MPC learns to achieve >**40%** better goal reached in highly constrained environments and >**65%** better behavior as captured by social metrics defined in the Appendix when moving around humans in a social navigation pilot study.

## 2 Performer-MPC: Learnable MPC with Scalable Real-Time Attention

In order to respond to dynamic uncertainty in the environment, the principal challenges of synthesis of model predictive controllers [74, 75, 3] are: (i) to construct cost functions that remain suitable across a wide variety of robot-environment situations, and (ii) to generate reliable solutions to the underlying trajectory optimization problems in real-time. This work focuses on the first challenge above, in particular, by eschewing the classical approach of hand-engineered cost functions, and adopting a learning-based inverse optimal control [76, 77, 78, 79, 80, 81] framework, where the sensory/visual context is used to induce *what MPC problem to solve in real-time* in order to generate actions. We first provide some details on training and inference of a learnable MPC framework, and then discuss the details of the learnable components; see Fig. 2 for an overview.

### 2.1 Learnable Model Predictive Control

Let $C_0$ denote the current "context", e.g., such as a list of tensors encoding for example RGB/D streams, force-torque readings, and proprioceptive states over a short time-window. Consider a

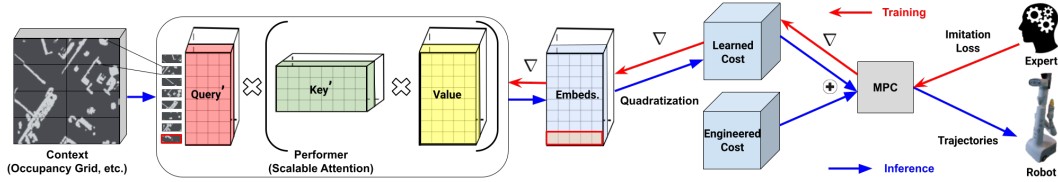

Figure 2: Overview of the Performer-MPC. The final latent embedding of the patch highlighted in red is used to construct context-dependent learnable cost. The backpropagation (red arrows) is through the parameters of the Transformer (see Sec. 2.2 for more details regarding Performers).

*Learnable-MPC* feedback policy implicitly defined by solving the following *parametric* optimal control problem at each time instant:

$$\arg\min_{\mathbf{u}_0 \ldots \mathbf{u}_{T-1}} \quad J_c(\mathbf{u}, \theta | C_0) = \sum_{t=0}^{T-1} c(\mathbf{x}_t, \mathbf{u}_t, t; C_0, \theta) + c_T(\mathbf{x}_T; C_0, \theta), \tag{1a}$$

$$\text{s.t.} \qquad \mathbf{x}_{t+1} = f(\mathbf{x}_t, \mathbf{u}_t; \theta), \qquad \mathbf{x}_0 = g(C_0, \theta) \text{ given}. \tag{1b}$$

Denote the optimizer as $\mathbf{u}^*(\mathbf{x}_0; C_0, \theta)$, and the corresponding optimal state sequence as $\mathbf{x}^*(\mathbf{x}_0; C_0, \theta)$. Here, $\theta$ are learnable parameters for stagewise and terminal cost neural networks $\{c, c_T\}$, $f$ the dynamics function, and $g$ current state estimator. While our framework generalizes to learning cost, dynamics, and state estimators simultaneously, in this paper we focus on the inverse optimal control setting: We study how the multi-layer self-attention cores of Transformers may be embedded in the *cost networks* to handle sensor fusion *while retaining real-time speed expectations of MPC*. The dynamics function here corresponds to the differential drive dynamics of our robot.

While the MPC-structured policy can be trained using any flavor of Reinforcement Learning, real-world trial-and-error data is prohibitively expensive and a general reward function that captures all intricacies of different real-world scenarios is difficult to design. Therefore, we take an Imitation Learning approach where the robot has access to $N$ expert demonstrations. The MPC structure provides a form of a strong inductive bias for Imitation Learning, and can lead to improved data efficiency, robustness, and generalization. Denote $\bar{\mathbf{u}}^i = (\bar{\mathbf{u}}_0^i, \ldots, \bar{\mathbf{u}}_{T_i-1}^i)$ and $\bar{\mathbf{x}}^i = (\bar{\mathbf{x}}_0^i, \ldots, \bar{\mathbf{x}}_{T_i}^i)$ as the control and state sequence for the $i^{th}$ demonstration snippet, with associated sensor context $C_0^i$, which can be extracted from offline planning or human teleoperation. We optimize $\theta$ as follows:

$$\theta^\star \quad = \quad \arg\min_\theta \sum_{i=1}^N J_l(\mathbf{u}^{i*}(\theta) | \bar{\mathbf{u}}^i, \bar{\mathbf{x}}^i), \tag{2}$$

where $J_l$ denotes total imitation loss that measures discrepancy between MPC-generated and expert state-control trajectories. We assume that $J_l$ also admits a stagewise and terminal decomposition using loss functions $l$ and $l_T$:

$$J_l(\mathbf{u}^{i*}(\theta) | \bar{\mathbf{u}}^i, \bar{\mathbf{x}}^i) \quad = \quad \sum_{t=0}^{T_i-1} l(\mathbf{x}_t^{i*}(\theta), \mathbf{u}_t^{i*}(\theta), t, \bar{\mathbf{x}}_t^i, \bar{\mathbf{u}}_t^i) + l_T(\mathbf{x}_{T_i}^{i*}(\theta), \bar{\mathbf{x}}_{T_i}^i), \tag{3a}$$

$$\mathbf{x}_{t+1}^{i*}(\theta) = f(\mathbf{x}_t^{i*}(\theta), \mathbf{u}_t^{i*}(\theta)), \qquad \mathbf{x}_0^{i*}(\theta) = \bar{\mathbf{x}}_0^i. \tag{3b}$$

Above, $\mathbf{x}^{i*}, \mathbf{u}^{i*}$ is the MPC solution with cost parameters $\theta$, given context $C_0^i$ and initial state $\bar{\mathbf{x}}_0^i$.

**Training via Bilevel Optimization:** The training optimization problem in (3) has embedded the MPC optimization, (1). Together, the two may be viewed as an instance of *bilevel optimization*, where the higher-level searches for the best cost-network parameters $\theta$ via imitation loss minimization, while the lower-level synthesizes the optimal predictive control sequences given fixed $\theta$. To use stochastic gradient descent for the higher-level problem, we need the gradient of $J_l$ with respect to $\theta$ evaluated at a control sequence $\mathbf{u}^*(\theta_k)$ where $\theta_k$ denotes the parameters during the current iterate $k$ during training. This quantity decomposes as a vector-Jacobian product (VJP),

$$\nabla_\theta J_l(\mathbf{u}^*(\theta_k) | \bar{\mathbf{u}}^i, \bar{\mathbf{x}}^i) = \nabla_\mathbf{u} J_l(\mathbf{u}^*(\theta_k) | \bar{\mathbf{u}}^i, \bar{\mathbf{x}}^i)^T \partial_\theta \mathbf{u}^*(\theta_k). \tag{4}$$

The first term in the product on the right hand side is the gradient of the total imitation loss, which can be efficiently computed using the Adjoint method in Optimal Control [82] thanks to its stagewise

structure. The second term is the sensitivity of the MPC solution with respect to parameters, which may be efficiently computed using the Implicit Function Theorem (IFT), as featured in several works exploring differentiable "optimization layers" [83, 84, 85, 86], see Appendix for details.

**MPC Solver:**    We use a second order Gauss-Newton trajectory optimizer [87, 88] called Iterative LQR (iLQR) [89] with line searches inspired by Differential Dynamic Programming (DDP) [90, 87]. At each iteration, iLQR quadratizes the cost and linearizes the dynamics to compute the search direction by solving a time-varying LQR (TVLQR) [82] problem. Upon convergence, a single additional LQR solve suffices for computing $\partial_\theta \mathbf{u}^*(\theta)$ in (4).

**Policy:**    While one may use the first component of the optimal solution for problem (1), i.e., $\mathbf{u}_0^*(\mathbf{x}_0; C_0, \theta)$, as the policy map, we noted better performance by leveraging a secondary (non-learnable) MPC problem similar to (1), featuring a "tracking" objective w.r.t. the solution of the learnable MPC problem. The details of this "tracking MPC" problem are provided in the Appendix.

## 2.2    Attentive Cost Functions for Learnable MPC

We adopt an inverse optimal control framework for learnable MPC whereby only the cost function is learnable. We structure this cost as the sum of a user-engineered function and a context-dependent quadratic, parameterized by an embedding matrix $\mathbf{P}$ and vector $\mathbf{q}$ (described in more detail below):

$$c(\mathbf{x}, \mathbf{u}, t; C_0, \theta) = \bar{c}(\mathbf{x}, \mathbf{u}, t) + \begin{bmatrix} \mathbf{x} \\ \mathbf{u} \end{bmatrix}^T \mathbf{P}^T(C_0, \theta) \mathbf{P}(C_0, \theta) \begin{bmatrix} \mathbf{x} \\ \mathbf{u} \end{bmatrix} + \mathbf{q}(C_0, \theta)^T \begin{bmatrix} \mathbf{x} \\ \mathbf{u} \end{bmatrix}. \quad (5)$$

Here, $\bar{c}$ refers to the hand-designed cost function (see Appendix), appended to a Transformer-backed cost model that attends to the current context $C_0$ to generate residual quadratic cost terms for MPC to optimize. This structure removes the computational cost of repeated quadratization of a large network in the iLQR solver. Furthermore, since the residual cost is convex and well-conditioned, crude MPC solutions can generate reliable descent directions for the higher-level optimizer, even though applying IFT in gradient computation assumes the MPC solution is precisely a local minimum. We next describe the details of the embeddings $\mathbf{P}$ and $\mathbf{q}$.

**Generating Context-Dependent Transformer Embeddings:**    We outline a general Transformer-based backend for learnable-MPCs which leads to Performer-MPCs. The backend maps the current contexts $C_0$ into a latent embedding which can be reshaped into the matrices $\mathbf{P}$ and $\mathbf{q}$ to support the quadratic parameterization of Eqn. 5. For concreteness, let $C_0$ be an image frame, i.e., the occupancy grid in the robot frame. As in Vision Transformer architectures [91, 92, 93], each frame is first independently pre-processed by a convolution layer, and then flattened to a sequence. Each element (token) of the sequence corresponds to a different patch of the original frame which is then enriched with positional encodings. The length $L$ of this sequence is a patch-size hyper-parameter. The preprocessed input is then fed to regular attention and MLP layers. The final embedding of one of the tokens is chosen as a latent representation of the entire context $C_0$ to parameterize the learnable cost (e.g., via de-vectorization to $\mathbf{P}$ and $\mathbf{q}$ as in Eqn. 5). Even though we take the final embedding of a single token, it contains signal from all the tokens since attention mixes information across tokens.

The attention used in regular Transformer architectures [52] linearly projects tokens' embeddings (via trainable transformations) into three matrices, $\mathbf{Q}, \mathbf{K}, \mathbf{V} \in \mathbb{R}^{L \times d}$, called *queries*, *keys* and *values* respectively. The output of the attention is then defined as:

$$\text{Att}(\mathbf{Q}, \mathbf{K}, \mathbf{V}) = \mathbf{D}^{-1} \mathbf{A} \mathbf{V}, \quad \mathbf{A} = \exp(\mathbf{Q}\mathbf{K}^\top / \sqrt{d}), \quad \mathbf{D} = \text{diag}(\mathbf{A}\mathbf{1}_L), \quad (6)$$

where $\mathbf{A} \in \mathbb{R}^{L \times L}$ is called the *attention matrix*. In the above formula, $\exp(\cdot)$ is applied element-wise, $\mathbf{1}_L$ is the all-ones vector of length $L$, and $\text{diag}(\cdot)$ is a diagonal matrix with the input vector as the diagonal. Space and time complexity of computing Eqn. 6 are: $O(L^2 + Ld)$ and $O(L^2 d)$ respectively, because $\mathbf{A}$ has to be explicitly stored. Quadratic time and space complexity in the number of patches makes this approach prohibitive for real-time robotic navigation. To address this, we apply a class of low-rank implicit linear-attention Transformer architectures, called *Performers* [73].

Performers interpret attention matrices $\mathbf{A} \in \mathbb{R}^{L \times L}$ as kernel matrices, i.e., $\mathbf{A}(i, j)$ is defined as: $\mathbf{A}(i, j) = \text{K}(\mathbf{q}_i, \mathbf{k}_j)$, with $\mathbf{q}_i / \mathbf{k}_j$ standing for the $i^{th}/j^{th}$ query/key row-vector in $\mathbf{Q}/\mathbf{K}$ and kernel

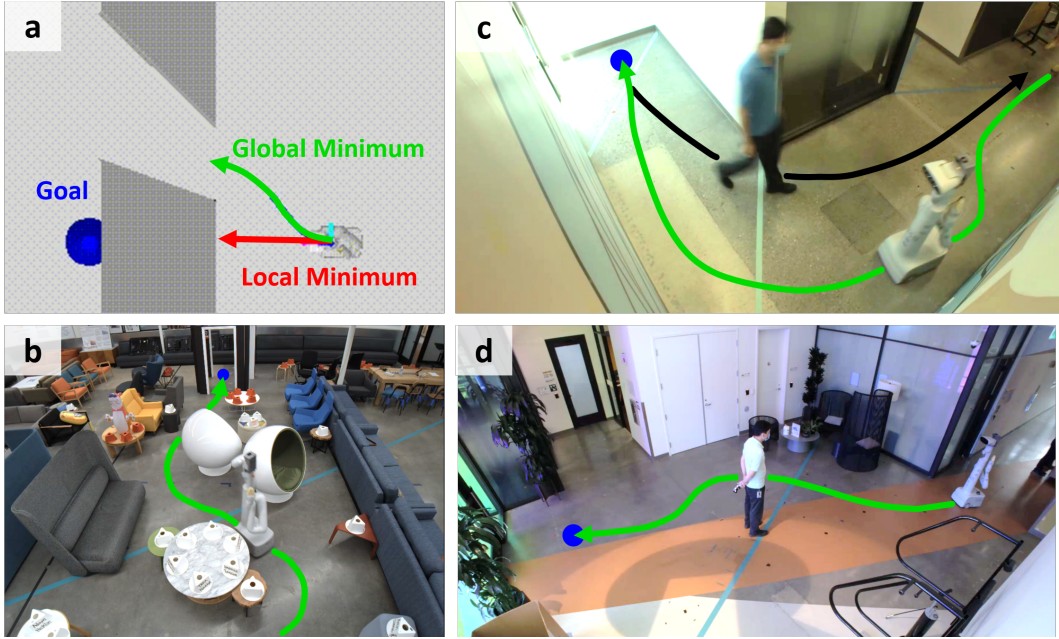

Figure 3: Experiment Scenarios: (a) Learning to avoid local minima during doorway traversal, (b) to maneuver through highly constrained spaces, and (c) to enable socially compliant behaviors for blind corner and (d) pedestrian obstruction interactions.

$\mathrm{K} : \mathbb{R}^d \times \mathbb{R}^d \to \mathbb{R}_+$ defined for the (randomized) mapping $\phi : \mathbb{R}^d \to \mathbb{R}^m$ (for $m > 0$) as:

$$\mathrm{K}(\mathbf{x}, \mathbf{y}) = \mathbb{E}[\phi(\mathbf{x})^\top \phi(\mathbf{y})]. \tag{7}$$

We call $\phi(\mathbf{v})$ a *(random) feature map* for $\mathbf{v} \in \mathbb{R}^d$. For $\mathbf{Q}', \mathbf{K}' \in \mathbb{R}^{L \times m}$ with rows given as $\phi(\mathbf{q}_i)$ and $\phi(\mathbf{k}_i)$ respectively, Eqn. 7 leads directly to the efficient attention mechanism of the form:

$$\widehat{\mathrm{Att}}(\mathbf{Q}, \mathbf{K}, \mathbf{V}) = \widehat{\mathbf{D}}^{-1}(\mathbf{Q}'((\mathbf{K}')^\top \mathbf{V})), \qquad \widehat{\mathbf{D}} = \mathrm{diag}(\mathbf{Q}'((\mathbf{K}')^\top \mathbf{1}_L)). \tag{8}$$

Here $\widehat{\mathrm{Att}}$ stands for the approximate attention and the parentheses indicate the order of computations. Such a mechanism is characterized by space complexity of $O(Lm + Ld + md)$ and time complexity of $O(Lmd)$ as opposed to $O(L^2 + Ld)$ and $O(L^2 d)$ of the regular attention mechanism. Different mappings $\phi$ give rise to different Performer variants. Two most popular ones [73] are:

$$\begin{aligned}
\phi_{\exp}(\mathbf{x}) &\overset{\mathrm{def}}{=} \frac{1}{\sqrt{m}} \exp(-\frac{\|\mathbf{x}\|_2^2}{2})\left(\exp(\omega_1^\top \mathbf{x}), \dots, \exp(\omega_m^\top \mathbf{x})\right) \text{ and,} \\
\phi_{\mathrm{relu}}(\mathbf{x}) &\overset{\mathrm{def}}{=} \frac{1}{\sqrt{m}}(\mathrm{ReLU}(x_1), \dots, \mathrm{ReLU}(x_d)) \text{ (with } m = d\text{)},
\end{aligned} \tag{9}$$

for $\omega_1, \dots, \omega_m \sim \mathcal{N}(0, \mathbf{I}_d)$ and we will refer to the corresponding Performers as *Performer-Exp* and *Performer-ReLU* respectively. Mapping $\phi_{\exp}$ provides unbiased estimation of the softmax attention-kernel from Eqn. 5, but requires random projections, whereas $\phi_{\mathrm{relu}}$ defines weaker attention-kernel, but does not use random projections and leads to the fastest Performer variant. In Section 3.2 and the Appendix, we provide a comprehensive speed benchmark for Performer-MPC variations, demonstrating Pixel-to-Pixel attention at real-time speeds.

## 3 Experiments

Our `Performer-MPC` is tested on a differential-drive wheeled robot, which has a 3D LiDAR in the front, and depth sensors mounted on its head (see Fig. 1). It is a three-layer Performer-ReLU model with $\mathrm{mlpdim} = 64$ and one head. Our policies are trained on four TPUs and then deployed on a CPU onboard the robot. We use the differentiable Iterative LQR implementation of `trajax` [94] for MPC training and inference. Please refer to the Appendix for more details.

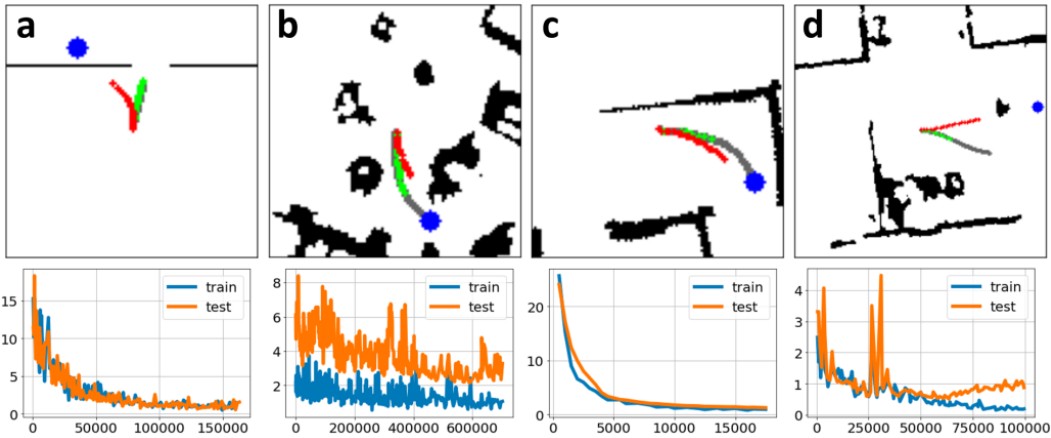

Figure 4: Top: Visualization of test data examples in the (a) doorway traversal, (b) highly constrained obstacle course, (c) blind corner, and (d) pedestrian obstruction scenarios. Performer-MPC trajectories aiming at the goal are always closer to the expert demonstrations compared to the RMPC trajectories. Bottom: Train and test curves, where the vertical axis represents loss values (Hausdorff distance to the expert trajectories) and horizontal axis represents training steps.

Performer-MPC is compared with two baselines, a regular MPC policy (RMPC) without the learned cost components, and an Explicit Policy (EP) that predicts a reference/goal state using the same Performer architecture, but without being coupled to the MPC structure. The control action (i.e., final policy output) is implicitly defined via the solution of the "tracking MPC" problem (see previous section), where the reference trajectory is generated by one of RMPC, EP, or Performer-MPC.

We evaluate our method in four scenarios, one in simulation and three in the real world (Fig. 3). For each scenario, the learned policies (EP and Performer-MPC ) are trained with demonstrations specifically collected for that scenario. To address the distribution shift issue, we not only collect positive examples where the robot is driving smoothly with the intended behavior, but also start the robot in randomly selected "disadvantage locations" (e.g., near-collision situations), and steer the robot to recover from them. For more data collection details please refer to the Appendix. We visualize the planning results of Performer-MPC (green) and RMPC (red) along with expert demonstrations (grey) in the top half of Fig. 4 and the train and test curves in the bottom half.

### 3.1 Experimental Results

**Learning to Avoid Local Minima:** We first evaluate our method in a simulated doorway traversal scenario (Fig. 3a). 100 start and goal pairs are randomly sampled from opposing sides of the wall. A planner guided by a greedy cost function often leads the robot to a local minimum, i.e., getting stuck at the closest point to the goal on the other side of the wall. Although such a problem can be mitigated by using a global planner, we use this as a test case to showcase the learning results. We generate 2000 expert demonstrations using an off-line iLQR planner [89], which iteratively solves for intermediate way points provided by a Dijkstra's global planner. Using these off-line demonstrations, Performer-MPC learns a cost landscape that steers the robot towards the doorway, even if it must veer away from the goal and travel further. Performer-MPC passes the doorway in 86 out of 100 trials while RMPC only passes 24 out of 100.

**Learning Highly-Constrained Maneuvers:** We next test our method in a challenging real-world scenario—a cluttered home/office setting where the robot must perform sharp, near-collision maneuvers (Fig. 3b). A global planner provides coarse way points for the robot to follow. Each policy is run ten times and we report Success Rate (SR) and average Completion Percentage (CP) with variance (VAR) of the obstacle course that the robot is able to traverse without collisions or getting stuck (Fig. 5). Performer-MPC outperforms both RMPC and EP in SR and CP.

**Learning to Anticipate Pedestrians at Blind Corners:** Going beyond static obstacles, we apply our method to social robot navigation [1], where robots must respect unwritten social norms for which cost functions are hard to design. One such scenario is blind corner (Fig. 3c), where robots

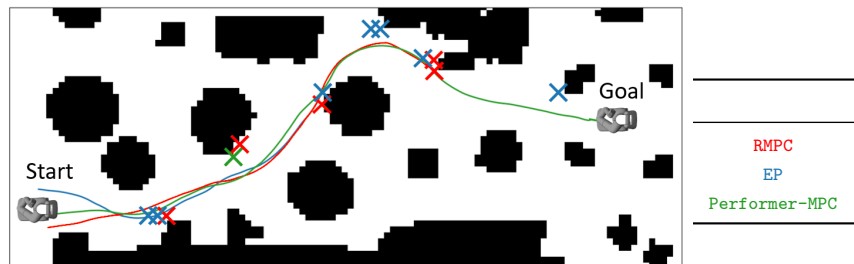

| | SR | CP ± VAR |
|---|---|---|
| RMPC | 5/10 | $69 \pm 13\%$ |
| EP | 3/10 | $62 \pm 15\%$ |
| Performer-MPC | **9/10** | $\mathbf{92 \pm 6\%}$ |

Figure 5: A $4.5 \times 10$ m$^2$ obstacle course with policy trajectories and failure locations indicated by crosses for RMPC, EP, and Performer-MPC.

should avoid the inner side of a hallway corner in case a human suddenly appears in this "blind spot". For blind corner, we collect 30 demonstrations with a human driving the robot from a randomly chosen location on one side of the corner to the other side in a socially compliant manner. After training, we evaluate each policy twenty times in the real world: ten times in the corner where the demonstrations were collected (seen), and ten times in a different corner (unseen). During each run, the robot and a pedestrian human subject will approach the corner from opposite sides, while a third-party observer monitors from a distance. We use a social navigation evaluation protocol [95] in which pedestrians and observers rate the performance of the policy using a standardized questionnaire scored with Likert scales [96] which we combine into a joint social navigation score (see Appendix for further details). Social navigation scores for blind corner seen and unseen scenarios are shown in Fig. 6a. RMPC has the least social compliance: its hand-crafted cost function efficiently cuts the corner, causing uncomfortable near-collisions (Fig. 1). EP performs slightly better than Performer-MPC in the seen environments, but does not generalize well to unseen scenarios, with worse social scores and a $20\%$ failure rate (e.g., safety stops or not reaching the goal).

**Learning to Respect Comfort Distance When Obstructed by Pedestrians:** Another common social navigation scenario is pedestrian obstruction, when a human unexpectedly impedes the prescribed path of a robot (Fig. 3d). While static obstacle avoidance is a largely solved problem, pedestrian obstruction is particularly challenging for MPC policies that are guided by way-points that were valid before the human entered the environment. A hand-crafted cost function may guide the robot too near to the human, causing uncomfortably close interactions or the robot getting stuck right in front of the human. We evaluate policy performance for pedestrian obstruction using the social navigation evaluation protocol [95]. Again, Performer-MPC is the most socially compliant (Fig. 6b), and in a few cases even shows emergent maneuvers unseen in the dataset (i.e., passing the human on the left if there is not enough space on the right, while the demonstrations only include right-side passing). In contrast, RMPC usually gets stuck in front of the human due to a local minimum close to the goal behind the human. While EP does stay away from the human

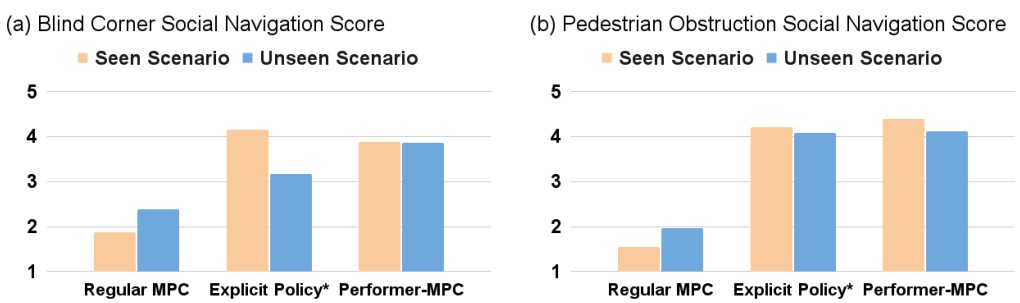

Figure 6: Social Navigation Results. (a) In the blind corner scenario, Performer-MPC achieves a better social navigation score than RMPC and similar to EP in the seen environment. In the unseen environment Performer-MPC is better than both baselines. EP fails to complete unseen $20\%$ of the time (e.g., not reaching the goal or triggering a safety stop, denoted by *). (b) In the pedestrian obstruction scenario, both EP and Performer-MPC show similar social scores and are superior to RMPC, but EP fails in seen or unseen conditions $5\%$ of the time (again denoted by *). Performer-MPC thus shows better generalizability in both social score and navigation success.

subject in both `seen` and `unseen`, it struggles to reach the goal location and sometimes comes close to colliding with nearby walls, leading to a $5\%$ overall failure rate. Please refer to the Appendix for a detailed description of both social scenarios and statistical analysis of the results.

## 3.2 Speed Studies Over Various Performer Architectures

Below we present speed ablation studies over various Performer architectures leading to Performer-ReLU as a default choice. More detailed studies are given in the Appendix. Tests presented here are run on $100 \times 100$ images, with patch size $5 \times 5$ and two architecture sizes: (a) **medium** with $l = 3$ layers, $h = 1$ head, and $\text{mlpdim} = 64$ (8.3M parameters) and (b) **large** with $l = 6$ layers, $h = 3$ heads, and $\text{mlpdim} = 1024$ (24.6M parameters). We benchmark Performer-ReLU and Performer-Exp (for the latter one varying the number of random projections rps used and either redrawing them or not at each forward pass), measuring wall-clock time taken by the MPC. For the fastest Performer-ReLU variant, we run additional studies (this time measuring CPU-time to distill time taken solely by the MPC from I/O time, etc.) for varying patch sizes, showing that *we can reach Pixel-to-Pixel attention with near real-time speed* (11.3ms per MPC iteration). The results are presented in tables Tab. 1 and 2.

Table 1: Speed ablation tests for different variants of Performers with $100 \times 100$ input images. The architecture deployed on the real robot is denoted in **bold**.

| model size | Performer type | redraw | rps # | wall-clock time per MPC-iteration [ms] |
|---|---|---|---|---|
| **medium** | **ReLU** | **N/A** | **N/A** | **8.3** |
| medium | Exp | False | 8 | 21.5 |
| medium | Exp | False | 16 | 21.6 |
| medium | Exp | False | 64 | 23.6 |
| large | ReLU | N/A | N/A | 13.8 |
| large | Exp | False | 8 | 72.9 |
| large | Exp | False | 16 | 66.9 |
| large | Exp | False | 64 | 83.0 |

Table 2: Speed ablation over patch sizes for medium-sized Performer-ReLU with $100 \times 100$ inputs.

| patch sizes | $1 \times 1$ | $2 \times 2$ | $4 \times 4$ | $5 \times 5$ | $10 \times 10$ |
|---|---|---|---|---|---|
| number of params (M) | 15.7 | 9.93 | 8.50 | 8.33 | 8.16 |
| CPU-time per MPC-iteration (ms) | 11.3 | 2.6 | 2.3 | 1.5 | 0.5 |

## 4 Limitations

Currently our Transformer-backend uses spatial attention, but in principle it can leverage the temporal axis. For example, in a face-to-face approach with a walking human in a hallway, motion history may shed light on how the human intends to move in the future, e.g., yielding left or right. A promising future research direction is to add history dependency so that the robot can sequentially reason about potential future interactions and conflicts. Furthermore, exploring richer modalities [97] than the occupancy grid (e.g., RGB images, human traces, language contexts [98, 72]), to enable robot-environment and robot-human interactions beyond simple geometry is another natural way to extend our approach. Another limitation is while the quadratic cost assures global convexity and training stability, it limits the expressiveness and complexity of the cost function. Furthermore, the cost function is learned individually for each navigation scenario, but it is unclear how one stand-alone learned cost function can handle multiple scenarios. Also, our user study pilot questionnaire could be refined, and our social evaluations could be expanded to a broader set of scenarios.

## 5 Conclusions

We present in this paper Performer-MPC, a learnable MPC system utilizing scalable Transformers to learn rich context representations parameterizing trainable cost function. We show that Performer-MPCs can be used as robotic controllers for navigation in challenging real-world environments where regular MPCs struggle, including learning to avoid local minima, to maneuver through highly constrained spaces, and to adhere to unwritten social norms, while maintaining real-time speed, even for nearly Pixel-to-Pixel attention.

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
