# OpenReview forum: "Learning Model Predictive Controllers with Real-Time Attention for Real-World Navigation"
_robot-learning.org/CoRL/2022/Conference — CoRL 2022 Poster_

### Official Review · Reviewer_Xcqt · 2022-07-21

**Originality:** Good
**Technical Quality:** Fair
**Clarity Of Presentation:** Very Good
**Impact:** 2

**Recommendation:**

Weak Reject: I recommend rejecting the paper, but will not argue for my recommendation if the majority of other reviewers have a different opinion.

**Summary:**

This paper presents a novel learning-based MPC approach where the cost function is learned using the expressiveness of deep neural networks such as transformers. These extract the cost directly from context information and trained using imitation learning while the MPC is solved using classic optimization techniques. The proposed solution is compared with two main baselines. The first one includes is a classic MPC whereas the second one is an explicit policy method that directly generates control actions without using the MPC structure.

**Issues:**

As explained in the other section there are several issues that have been summarized.

**Quality Of The Limitations Section:**

Limitations are addressed clearly

**Reviewer Expertise:**

5: The reviewer is absolutely certain that the evaluation is correct and very familiar with the relevant literature

**Robotics Focus:**

Sufficient demonstration on hardware

**Strengths And Weaknesses:**

The paper is well written and the approach is interesting. However, there are several aspects that are not fully convincing. First, it is not clear why the proposed representation as input to generate the final latent embedding is selected. How other representations affect the learning module. This is not clear. Second, the current setup is tested and analyzed in separate scenarios and every time needs to be separately retrained. This is a huge limitation of this approach and it is not clear if the architecture is able to learn different scenarios at the same time. The network is tested except for minor variation in the same scenario or task it was trained for. Third, it is not clear by using a Performer type of architecture rather than the original transformer if there is any loss in performance or in the ability to learn the cost function. It would have been interesting to have some study about that at least showing what the different architectures learn in terms of costs if not possible to deploy all of them directly in the MPC for computational reasons. This can also be extended considering different architectures beyond transformers. Finally, it is not clear what is the model used in the current robot and at which level of abstraction the control commands are sent and how some constraints like pedestrian detections are incorporated in the regular MPC approach. An appropriate constraint should not let the robot get stuck in front of the human. The proposed comparison is probably not fair in this case.

**Summary Of Recommendation:**

The approach is interesting, but the there are several limitations that have been highlighted that prevent its ability to be fully utilized in real-world settings.

---

> ### Author Response · Authors · 2022-08-25
> **Follow up**
>
> Please kindly let us know if there remain any more concerns you would like us to address; Otherwise, would you please consider raising the evaluation score ? Thanks !

---

### Official Review · Reviewer_UiTD · 2022-07-27

**Originality:** Very Good
**Technical Quality:** Very Good
**Clarity Of Presentation:** Excellent
**Impact:** 4

**Recommendation:**

Strong Accept: I recommend accepting the paper and will argue for my recommendation even if other reviewers hold a different opinion.

**Summary:**

This paper presents a novel way of augmenting the cost function of a Model Predictive Controller by allowing the cost function to be a combination of a fixed one and a learned one. The learned part of the cost function takes the most general form of non-linear least squares and learns the matrix coefficients directly using the embeddings of a vision transformer architecture. They use Performers instead of Transformers to speed up the training and forward pass. Using bi-level optimization, the gradients are propagated through the MPC optimizer.

**Issues:**

 - Have the authors tried not to use a hand-designed cost function? There is no mention as to why the authors need to use this hand-designed cost function. It would be worth commenting about this.
 - It would be very nice to have (or comment on) an extra comparison against end-to-end policies trained using imitation learning in order to understand better the benefits of using an MPC in the loop.
 - Throughout the paper, it’s mentioned that the “Tracking MPC” is used for all the approaches, and in the Appendix, this decision is argued by stating that the comparisons with the other baselines are fairer because all approaches use it. But then, the entire approach is doing planning and not control. The tackled problem is the generation of a trajectory that is then tracked with an MPC. I believe that this fact deserves more explanations.
 - “Real-Time Attention” is in the title of the manuscript, and the application of “pixel-to-pixel attention mechanisms in MPC” is one of your contributions. However, there are no ablations regarding the attention mask. How are the attention mechanisms helping here? It would be interesting to see what the agent pays attention to in different scenarios, e.g., when avoiding a human or when going through the door.
 - The approach assumes knowledge of the state of the robot, but this fact is only mentioned in the appendix (we get the state of the robot from a SLAM pipeline). It would be desired to highlight this fact earlier in the manuscript since it’s an important aspect of the approach.


**Quality Of The Limitations Section:**

Limitations are addressed clearly

**Reviewer Expertise:**

4: The reviewer is confident but not absolutely certain that the evaluation is correct

**Robotics Focus:**

Sufficient demonstration on hardware

**Strengths And Weaknesses:**

The main strengths and learnings from this paper are:
 - The paper is very well written and very self-explanatory. The figures are very well chosen, and the theoretical part is rigorous and sound.
 - It’s the first approach (to my knowledge) that uses vision transformers in combination with an MPC in the loop to learn the cost function of said MPC. It can be an inspiration for future researchers, and it has the potential of being, in my opinion, a very good research direction.
 - It uses bi-level optimization to propagate the gradients through an optimizer. I believe that using this tool is a very elegant way of getting the best of both worlds, learning-based and model-based, to work together in what they respectively excel at.

And weaknesses:
 - While the approach renders very good results and it’s elegant, it is compared with rather naive baselines. Furthermore, the existence of the MPC in the loop is not ablated in any way (since in all approaches there is a “Tracking MPC” problem that is solved at the end).







**Summary Of Recommendation:**

I believe this paper can have a major impact on the way that Model Predictive Control is used together with learning-based approaches. The tendency to use control in the last layer in robotics applications is gaining more and more popularity, and I believe that it can be a natural way of including model-based guarantees in an otherwise learning-based architecture.

---

### Official Review · Reviewer_yaP4 · 2022-07-31

**Originality:** Good
**Technical Quality:** Very Good
**Clarity Of Presentation:** Good
**Impact:** 3

**Recommendation:**

Weak Accept: I recommend accepting the paper, but will not argue for my recommendation if the majority of other reviewers have a different opinion.

**Summary:**

The paper proposes an approach that allows incorporation of complex rules (such as social norms, obstacle avoidance in cluttered scenarios) into the behavior generated by standard MPC policies. The main idea of the approach is based around transformer-generated cost functions trained through imitation learning (behavior cloning in this instance). The authors demonstrate effectiveness of their approach on a real robotics platform in multiple scenarios, including navigation in cluttered scenes and socially-acceptable navigation in human-occupied environments.

**Issues:**

See my CONCERNS section

**Quality Of The Limitations Section:**

Limitations are addressed clearly

**Reviewer Expertise:**

3: The reviewer is fairly confident that the evaluation is correct

**Robotics Focus:**

Sufficient demonstration on hardware

**Strengths And Weaknesses:**

Praise:
* The work tackles an important problem of incorporating complex behaviors, while retaining advantages of the existing control approaches: optimality, interpretability.
* The work conducts interesting experiments on a real robotics platform and demonstrates some level of generalization on novel (albeit similar) cases.


Concerns:
* Literature overview is not complete: Authors cite a lot of papers for transformers in unrelated/weakly-related fields (which is definitely excessive), but the section on inverse-RL and cost learning is tiny. Even the relevant papers cited by the authors are just mentioned: there is no detailed comparison, i.e. how the current work advances already existing methods? It makes it harder to appreciate what has been done in the paper. Just to name a few other examples from the existing relevant body of literature that is worth including / comparing to:
   * Sermanet et al. Time-Contrastive Networks: Self-Supervised Learning from Video.
   * Hsu et al. Unsupervised Learning Via Meta-Learning
   * Finn et al. Guided Cost Learning: Deep Inverse Optimal Control via Policy Optimization.
   * Bechtle et al. Meta-Learning via Learned Loss.

   * Each task is learned separately, hence, it brings a question how to stitch them later. Although the authors explicitly mention this limitation it renders results quite a bit less impressive. Also, the experiments are conducted in somewhat similar environments and do not exhibit strong distributional shifts under proposed inputs.

**Summary Of Recommendation:**

The work tackles an important problem and demonstrates interesting results. The paper is decently written and generally easy to follow.
Among the issues are i) somewhat weak comparison to the existing body of work and ii) overall results could demonstrate better generalization under multiple tasks and stronger distributional shifts.
The ideas and results are not revolutionary by any means, but I appreciate the amount of work put into producing this paper - it has decent quality and results look convincing, hence I am inclined to accept it.

---

> ### Author Response · Authors · 2022-08-25
> **Follow up**
>
> Please kindly let us know if there remain any more concerns you would like us to address; Otherwise, would you please consider raising the evaluation score ? Thanks !

---

### Official Review · Reviewer_3ALJ · 2022-08-01

**Originality:** Very Good
**Technical Quality:** Good
**Clarity Of Presentation:** Excellent
**Impact:** 4

**Recommendation:**

Weak Accept: I recommend accepting the paper, but will not argue for my recommendation if the majority of other reviewers have a different opinion.

**Summary:**

This paper introduces a novel learnable MPC approach by enhancing the traditional MPC cost with learnable components. These learnable components use transformer-based embeddings of real-world context. The overall model is learned in imitation learning fashion by leveraging expert demonstrations. Various optimizations are considered to enable the model to run in real-time. Through experiments, the model is less likely to be stuck in local optimum and are more social when compared to the traditional MPC. The model is also less likely to run into obstacles when compared to pure expert demonstration models.

**Issues:**

Copied from strengths and weaknesses:
- For learning highly constrained maneuvers, are the human expert demonstrations conducted in the same cluttered environment as in the evaluation? How well does Performer-MPC perform in unseen cluttered environments when compared to RMPC. (Or how does RMPC perform because I saw Performer-MPC generalizes in one instance shown in the video)
- For blind corner and pedestrian obstruction scenarios, I do not see any statistical analysis. Are the comparisons statistically significant among the social score ratings of RMPC, EP and Performer-MPC? How many subjects are there? Are these the same as the pilot studies mentioned in the supplementary materials?

The reference section may need some organizing: For example, I saw a double reference [1] and [48].

**Quality Of The Limitations Section:**

Additional details required

**Reviewer Expertise:**

4: The reviewer is confident but not absolutely certain that the evaluation is correct

**Robotics Focus:**

Sufficient demonstration on hardware

**Strengths And Weaknesses:**

This paper is very clearly written. The authors have very cleanly formulated their approach and provide good reasoning describing every single step. I also concur that learnable-MPC combines the best of both worlds with the learning part being able to encode complex environmental and social context and the MPC part providing safety guarantees to some extent. The experiments are also carefully designed around 4 example scenarios and support the authors' claims.

I still have a few questions about the experiments:
- For learning highly constrained maneuvers, are the human expert demonstrations conducted in the same cluttered environment as in the evaluation? How well does Performer-MPC perform in unseen cluttered environments when compared to RMPC. (Or how does RMPC perform because I saw Performer-MPC generalizes in one instance shown in the video)
- For blind corner and pedestrian obstruction scenarios, I do not see any statistical analysis. Are the comparisons statistically significant among the social score ratings of RMPC, EP and Performer-MPC? How many subjects are there? Are these the same as the pilot studies mentioned in the supplementary materials?

I am also curious about how this method generalizes to more complex pedestrian scenarios, such as navigating through a crowd. How many demonstrations are needed to be successful in this? Crowd behaviors are highly variable and unconstrained.

**Summary Of Recommendation:**

The paper's approach is novel and can have a very good impact on the research community. Its experiment designs are pretty thorough, but a few questions remain. My suspicions are RMPC might also work well in unseen cluttered environments. And Performer-MPC may not be statistically better than EP when it comes to social score ratings.

---

> ### Author Response · Authors · 2022-08-25
> **Follow up**
>
> Please kindly let us know if there remain any more concerns you would like us to address; Otherwise, would you please consider raising the evaluation score ? Thanks !

---

### Meta-Review · Area_Chair_C9gn · 2022-08-09

**Recommendation:** Accept (Poster)
**Confidence:** 4

**Metareview:**

This paper proposes a method called performer-MPC, which uses a transformer-based learned cost function by imitation learning with visual input to be combined with MPC for real-world robot navigation tasks. Its effectiveness is demonstrated in multiple real-world navigation scenarios and confirmed performance improvement over standard MPC.

Strengths:

- The paper is well written

- Strong technical contribution in bi-level optimization

- Real-world experiments with four scenarios that support the authors’ claims

Weaknesses:

- Lack of some details in the proposed method, experimental setups, and results

- Literature review is not sufficient

- No comparisons with SOTA

----- Post rebuttal -----

In the rebuttal period, some reviewers’ concerns were resolved and turned positive, and all the reviewers agreed that this paper should be accepted. Therefore, I conclude that the paper should be accepted. I strongly encourage the authors to take all the reviewers’ suggestions for the final version of the paper.



**Best Paper Nomination:**

No